# The Congestion “Pandemic” in Acute Heart Failure Patients

**DOI:** 10.3390/biomedicines12050951

**Published:** 2024-04-24

**Authors:** Daniela Mocan, Radu Ioan Lala, Maria Puschita, Luminita Pilat, Dan Alexandru Darabantiu, Adina Pop-Moldovan

**Affiliations:** 1Multidisciplinary Doctoral School, Vasile Goldis Western University of Arad, 310025 Arad, Romania; hoza-mocan.daniela@student.uvvg.ro (D.M.);; 2Cardiology Department, Arad County Clinical Emergency Hospital, 310037 Arad, Romania

**Keywords:** heart failure, congestion, volume overload, volume redistribution, biomarkers

## Abstract

Congestion not only represents a cardinal sign of heart failure (HF) but is also now recognized as the primary cause of hospital admissions, rehospitalization, and mortality among patients with acute heart failure (AHF). Congestion can manifest through various HF phenotypes in acute settings: volume overload, volume redistribution, or both. Recognizing the congestion phenotype is paramount, as it implies different therapeutic strategies for decongestion. Among patients with AHF, achieving complete decongestion is challenging, as more than half still experience residual congestion at discharge. Residual congestion is one of the strongest predictors of future cardiovascular events and poor outcomes. Through this review, we try to provide a better understanding of the congestion phenomenon among patients with AHF by highlighting insights into the pathophysiological mechanisms behind congestion and new diagnostic and management tools to achieve and maintain efficient decongestion.

## 1. Introduction

Heart failure (HF) has become a profound global health concern, attracting attention due to its widespread impact and alarming mortality rates. With an estimated 64.3 million individuals affected worldwide and one-year mortality rates ranging between 21% and 36%, HF poses a significant challenge in the realm of public health. As communities grapple with the aftermath of previous health crises, HF underscores the urgent need for a comprehensive understanding and proactive measures to address this escalating global-scale issue [1]. The new universal definition of HF considers this condition a clinical syndrome with current/prior symptoms and signs caused by a structural or functional cardiac abnormality, corroborated by elevated natriuretic peptide levels and evidence of pulmonary or systemic congestion [2].

Congestion encompasses more than just clinical signs and symptoms, and it appears to be the critical player underlying the complex pathophysiology of HF [3].

Various factors determine the clinical course of congestion among patients with acute heart failure (AHF), including kidney impairment, volume overload or redistribution, diuretic resistance, electrolyte disorders, low blood pressure, and inflammation. All these factors can lead to incomplete decongestion at discharge, thus influencing the prognosis regarding future rehospitalizations and mortality. In this review, by immersing ourselves in the pathophysiological mechanisms of congestion, unraveling its phenotypes, and discussing emerging diagnostic tools for targeting decongestive therapy, we aim to acknowledge the importance of ‘the congestion phenomenon’ and its scale among patients with HF, which is of pandemic proportions.

## 2. Understanding Congestion: Volume Overload or Volume Redistribution

Congestion is manifested clinically through signs and symptoms due to extracellular fluid accumulation due to increased left-sided cardiac filling pressures [3,4]. The increase in cardiac filling pressures is an early indicator of hemodynamic congestion, which precedes the development of congestive symptoms in days or weeks [5]. Congestion leads to HF decompensation, which manifests clinically with dyspnea, orthopnea, systemic edema, jugular venous distention, and third heart sound [3,5]. It is important to understand the underlying mechanism that first leads to hemodynamic congestion and then to clinical congestion, so early recognition and proper treatment of this condition are imperative. One must first identify the transition phase between hemodynamic and clinical congestion to do so.

Congestion results from the combined effects of the forward and backward failure of the heart, coupled with the inadequacy of compensatory adaptive mechanisms to counter the detrimental impact of reduced oxygen delivery to peripheral tissues. Reduced cardiac output, as well as systemic venous congestion, results in renal hypoperfusion, thus leading to neurohormonal activation with sodium (Na) and water accumulation and consequent volume overload [6]. Na is stored in the extracellular compartment, mainly in the interstitium at 65% and 25% in the intravascular compartment [7]. Congestion is the cardinal manifestation of both chronic heart failure (CHF) and AHF. Figure 1 illustrates volume redistribution and volume overload as the primary mechanisms underlying venous congestion and the coexistence of these two phenomena.

### 2.1. Volume Redistribution

The first mechanism involves the movement of blood volume from systemic circulation to pulmonary venous circulation. This phenomenon is attributed to a decrease in the capacity of the splanchnic venous bed, prompting expedited blood redistribution [8]. Volume redistribution was assessed in three major trials, COMPASS-HF, HOMEOSTASIS–HF, and CHAMPION, by the invasive monitoring of hemodynamics. All findings consistently indicate that weight gain resulting from volume overload does not typically precede an episode of acute decompensation in most patients; instead, there is an observed increase in cardiac filling pressures [9,10,11]. HF is a state of neurohormonal activation, inflammation, and endothelial dysfunction that promotes veno-arterial constriction. This leads to a subsequent decrease in venous bed capacitance with consequent blood volume redistribution and increased preload and afterload [12]. Fluid shifts between the interstitial and intravascular compartments occur weeks before the acute event, mostly without weight change. The hyperstimulation of the α1 and α2 adrenergic receptors in splanchnic veins characterizes the mechanism, increasing cardiac filling and capillary wedge pressures [12]. This relatively silent mechanism leads to intravascular volume expansion, thus preparing the ground for what would be the perfect storm. This is termed asymptomatic hemodynamic congestion. Increased cardiac filling pressures lead to myocardial insult, thus triggering the acute event by a rapid translocation of up to 1 L of fluid into the interstitial and then the alveolar space, resulting in worsening dyspnea and clinical congestion [13].

### 2.2. Volume Overload

The second mechanism responsible for congestion is volume overload, a rather insidious process with an absolute increase in water and Na body content [14]. Volume overload is frequently observed in individuals with cardiorenal dysfunction and CHF [14]. Venous distension augments endothelial dysfunction and sympathetic activation, further promoting the fall in splanchnic venous capacitance and consequent fluid redistribution [15]. In other words, there is an overlap between these two mechanisms. Impaired renal Na excretion causes fluid retention [14]. Understanding the heart–kidney cross-talk as Na homeostasis, neurohormonal activation, and inflammation that impair nephron tubular flow, contributing to Na and water retention [16], this will lead to increased central venous and intra-abdominal pressure, further worsening renal function [17]. Most Na reabsorption takes place in the proximal renal tubule (65%), mediated by Na transporters and the sodium–potassium pump (Na/K ATP-ase) [18]. As Na is reabsorbed, water passively follows the osmotic gradient. This process is kept stable through glomerular–tubular feedback but becomes unstable in HF due to deleterious mechanisms. Increased peritubular oncotic pressure, renal venous pressures, and renal lymph flow promote Na reabsorption in heart failure (HF) in the proximal tubule [17]. In addition to this, every drop in the glomerular filtration rate (GFR) due to worsening renal function decreases the amount of urinary Na excretion. In HF, less water and solutes reach the loop of Henle due to increased reabsorption from the proximal tubule [17]. This, together with augmented Na reabsorption in the thick ascending part of the Henle loop, promoted by neurohormonal activation, leads to the incapacity of the kidneys to dilute urine and excrete free water [18,19]. At this stage, aldosterone levels remain high, and osmotic interstitial oncotic pressures additionally promote the reabsorption of Na and water, despite a prior reduction in Na delivery to the distal tubule [20].

### 2.3. Heart Failure Manifests a Strong Avidity for Sodium

Na is the key player responsible for fluid redistribution and overload [21]. This hypothesis has been observed in clinical studies, which showed that increased levels of total body Na are present in patients with HF, both with peripheral edema and without edema, before an acute event of decompensation [22,23]. Increased Na levels are associated with increased filling pressures. Recent evidence indicates that the glycosaminoglycan (GAG) network in the interstitium plays a highly regulatory role in fluid homeostasis by binding a large amount of Na, thereby functioning as a buffer for this electrolyte [24]. The GAG network has a stronger affinity for the Na cation than other ions and molecules, thus creating a hypertonic Na microenvironment [25]. Firstly, interstitial GAG bonds with excess Na molecules without causing subsequent water retention or altering Na plasma concentration [26]. The entrapped Na molecules in the interstitium matrix are practically deprived of interaction with vascular osmoreceptors, thus preventing the release of the arginine-vasopressin (AVP) hormone, which determines consequent water retention [24]. Additionally, once trapped, the interstitial Na cation evades the renal regulatory mechanism, significantly complicating its removal from the body [24]. This, so far, is a very potent explanation of why some patients with HF do not experience weight gain before acute decompensation. Eventually, due to high exposure to Na, the GAG network will lose its buffering capacity. The GAG architecture weakens and transforms, shifting from a low-compliance to a high-compliance compartment, thus facilitating fluid translocation [27]. In a dysfunctional GAG network, elevated venous pressures drive interstitial fluid transudation, surpassing the lymphatic drainage capacity and resulting in pulmonary and systemic congestion. [24]. However, an additional aspect to consider involves the endothelial glycocalyx, a vibrant glycoprotein network with vasoprotective functions [28,29]. It acts as a barrier against plasma, reduces vascular permeability, and prevents platelet and leukocyte adhesion, but most importantly, it acts as a Na buffer [29]. Mechanisms such as oxidative stress, ischemia, inflammation, excessive shear stress, increased Na concentration, and natriuretic peptides are hallmarks of HF and are responsible for endothelial glycocalyx disruption [30,31]. A damaged endothelial glycocalyx will increase vascular permeability and diminish Na buffering capacity [32]. The loss of its buffer capacity will expose the endothelial cells to many Na cations. This, together with high aldosterone concentrations, will stimulate the hyperactivation of the endothelial Na channels from the apical region of the endothelial cells, leading to high Na uptake [33]. Therefore, impaired smooth muscle cell contraction, decreased nitric oxide (NO) production, and endothelial stiffness will occur, augmenting the neurohormonal response and endothelial dysfunction [34].

The deterioration of the endothelial glycocalyx leads to elevated cardiac filling pressures and hemodynamic congestion, serving as the precursor to HF decompensations [21,24].

## 3. Does Congestion Matter in Heart Failure?

Congestion is the leading cause of HF hospitalization and readmission and is strongly associated with HF prognosis [5,35,36]. In extensive clinical trials, it has been observed that the predominant cause of hospital admissions in HF cases is attributable to manifestations of venous congestion rather than those indicative of low cardiac output.

There are distinct phases of congestion, encompassing hemodynamic, clinical, and systemic stages (Figure 2) [3]. The first stage is hemodynamic congestion, characterized by the elevation of cardiac filling pressures and the increase in venous pressures without clinical manifestation [37,38]. Subsequently, organ congestion is caused by the redistribution and accumulation of fluid within the extracellular and third space, and clinical congestion appears [37]. Hemodynamic congestion is responsible for HF progression and precedes acute HF decompensation episodes [39,40].

Furthermore, persistent hemodynamic congestion, despite symptom relief and aggressive diuretic therapy, is a prognostic marker for rehospitalization [39,40]. Ambrosy et al. confirmed this, showing that elevated natriuretic peptides before discharge reflect residual decongestion and are one of the strongest predictors of short-term outcomes [35]. Trials such as DOSE-AHF and CARESS-AHF have shown that nearly half of the patients hospitalized for decompensated HF are still not congestion-free at discharge and have higher readmission and mortality rates [41].

It is now recognized that a patient with HF is mainly exposed to adverse events, not during hospitalization but afterward, during the so-called “vulnerable phase” (VP) [42,43]. The VP follows an episode of acute HF exacerbation and lasts up to 6 months, during which patients carry a risk of readmission and mortality of 30% and 10%, respectively [42,43]. Although the factors responsible for the VP are numerous, one thing is sure. With each readmission, regardless of the precipitating factor involved in HF decompensation, there is a decline with further deterioration in cardiac function [44]. Of the numerous factors contributing to VP pathophysiology, one seems to weigh the most: failure to relieve congestion with the persistence of increased filling pressures, ultimately leading to hemodynamic congestion, clinical congestion, and multi-organ injury [45]. A summary of biomarkers with the ability to identify the population at the highest risk in this period was proposed, and these are natriuretic peptides, troponin, blood urea nitrogen (BUN), hematocrit, and serum osmolarity [5,43,45,46,47]. All these biomarkers, except troponin, point directly or indirectly to congestion and fluid retention. Having this in mind, tackling congestion as soon as possible is crucial. There is a need to find the most sensible and specific tools for accurately detecting congestion. The non-invasive assessment of congestion has been validated in preference for invasive assessment with different degrees of sensitivity and specificity [5]. The jugular venous pulse has the best sensitivity (70%) and specificity (79%) for detecting increased filling pressures and systemic congestion [4]. By using a simple composite congestion score that included dyspnea, orthopnea, fatigue, jugular venous distension (JVD), rales, and edema, Ambrosy et al. effectively demonstrated that a subset of HF patients remained with residual congestion before discharge [35]. This is how the EVEREST score was born and is considered to have the most evidence-based data regarding the congestion status of the patient with AHF [35].

## 4. Does the Heart Failure Phenotype Predict Congestion Mechanisms?

It has been suggested that the mechanism for the clinical manifestation of congestion (overload or redistribution) depends on the AHF phenotype. Diastolic dysfunction and elevated blood pressures characterize HF with preserved ejection fraction (HFpEF), making volume redistribution leading to pulmonary congestion the primary clinical manifestation of this phenotype rather than volume overload (Figure 3). In contrast, the phenotype of heart failure with reduced ejection fraction (HFrEF) is associated with volume overload, resulting in systemic congestion and weight gain, and is characterized by both systolic and diastolic dysfunction [48].

Van Aelst et al. addressed this issue by comparing HFpEF with HFrEF patients and showed no difference in the manifestation of congestion mechanisms between the groups [49]. The study highlights that fluid redistribution and accumulation coexist [49]. Indeed, the cardiorenal continuum and hypoalbuminemia are robust predictors of venous congestion in HF. However, these features are not exclusive to a particular HF phenotype, as they are present at similar rates across all HF phenotypes [50,51]. However, a missing link remains that should elucidate the diverse manifestations of congestion in all HF models, irrespective of ejection fraction. As is often the case in daily practice, the answer may be correct in front of us, and this is indeed the case with our elusive missing link concerning congestion manifestation. Reviewing the 1971 Framingham Criteria introduced by McKee et al., we note that jugular venous distension and hepato-jugular reflux were the primary criteria for defining congestive HF. These criteria exhibit high specificity for congestive HF and are also associated with elevated right atrial pressures and right ventricular (RV) dysfunction [52,53].

The elusive factor, consistently present and described across all HF phenotypes and capable of predicting congestion, may indeed be RV dysfunction [54]. It carries a poor prognosis in patients with HF, regardless of ejection fraction [54]. RV dysfunction/failure is responsible for increased central venous pressures and consequent systemic congestion. Still, recently, it has been pointed out that RV dysfunction and RV–pulmonary artery uncoupling are associated with pulmonary congestion [55]. Kobayashi et al. showed that in patients with acute decompensated HF, a low tricuspid annular plane systolic excursion (TAPSE) and TAPSE/PSAP (pulmonary artery systolic pressure) ratio correlate with an increased number of B lines detected by lung ultrasound upon admission and at discharge [55]. Other studies showed that chronic HF with RV dysfunction and RV-PA (pulmonary artery) uncoupling correlated with subclinical pulmonary and peripheral congestion and had a worse prognosis [56]. Elevated right atrial pressures might explain this by decreasing fluid drainage from the interstitial lung tissue via pulmonary lymphatics. Assessing RV function and the early recognition of RV dysfunction are crucial for managing patients with acute and chronic HF and thus improving the prognosis.

## 5. Clinical and Paraclinical Integrative Assessment of Congestion

### 5.1. Clinical Congestion Scores

Physical assessment is not accurate enough to detect low congestion levels. More than 50% of patients discharged from the hospital have residual congestion with high levels of natriuretic peptides [41].

Relying only on clinical signs has a low sensitivity and a poor predictive value for identifying decompensated HF, which is why congestion scores and biomarkers are essential tools [5,57].

Several scores have been tested and proposed to quantify congestion: the Lucas score, Gheorghiade score, Stevenson classification, Rhode score, and Everest score [5]. All of these scores combine several clinical and paraclinical indicators such as orthopnea, jugular venous distension, rales, edema, hepatomegaly, the third heart sound, the dose of diuretics, fatigue, orthostatic testing, 6 min walk test, and the N-terminal pro-brain natriuretic peptide (NT-proBNP) [35,57]. A brief overview of the scores is provided in Table 1. Although there is a high level of confidence in determining congestion by using these scores, there is still a lack of data for their use in routine clinical practice, as they serve so much more as a predictive tool than a management tool. However, emerging evidence points to the EVEREST score as a strong candidate for routine congestion management in acute HF [5,35].

### 5.2. The New Congestion Biomarkers on the Horizon

When it comes to biomarkers, natriuretic peptides have been extensively studied for congestion in HF, and their role in diagnosis and prognosis is well established in current guidelines [58]. However, their utility is limited because natriuretic peptides fail to demonstrate any advantage in guiding decongestion therapy compared to standard care. The situation becomes more intricate due to the potential elevation of natriuretic peptides in cases of ischemia or atrial fibrillation, which may not always indicate the presence of congestion [59,60,61]. Another pitfall is that natriuretic peptides fail to show the actual contribution of right-sided HF to the clinical congestion picture [62]. Thus, despite their evident utility, natriuretic peptides should be integrated with other clinical and paraclinical parameters to predict and manage congestion. A set of novel congestion biomarkers shows promise and merits consideration (Table 2).

An interesting congestion biomarker extensively studied recently in patients with congestive HF is carbohydrate antigen 125 (CA125), a glycoprotein expressed on serosal epithelial cells. Even though it is validated as a marker for ovarian cancer, it is highly expressed in conditions associated with volume overload, such as HF, renal failure, or liver failure [63]. Elevated hydrostatic pressure induces mechanical stress, leading to the overexpression of CA125 [64]. Several studies have demonstrated the association of CA125 with systemic congestion in patients with acute decompensated HF [62,64,65]. However, recent studies have revealed that the utility of CA125 extends beyond its role as a congestion marker. It has been demonstrated to predict both short- and long-term outcomes in decompensated HF, suggesting its potential to guide diuretic therapy [66,67]. However, some aspects regarding CA 125 use in HF need to be correctly comprehended.

Firstly, since its release is primarily associated with third-space fluid accumulation, it appears ineffective in identifying patients with acute-onset conditions characterized by predominant interstitial pulmonary congestion. Secondly, compared to natriuretic peptides, it has a longer half-life, up to 12 days, and confounders such as age or kidney function do not influence it [64]. On the other hand, these features present some advantages: CA 125 could detect patients with RV failure and chronic fluid overload, thereby impacting the escalation of decongestive therapy and the better management of HFpEF patients with associated comorbidities [64,68].

A new marker of congestion in HF that has recently emerged is CD146 (cluster of differentiation 146), a glycoprotein highly expressed at the junctions of endothelial cells throughout the human vascular system, smooth muscle cells, and pericytes [67]. It is an adhesion molecule active in venous integrity [68,69]. It is overexpressed and released into the bloodstream in conditions associated with endothelial dysfunction, vascular injury, or mechanical vascular stretch [70]. Different roles of CD146 have been described: angiogenesis, vessel permeability, and leukocyte transmigration [71,72]. Increased levels of CD146 have been linked to peripheral venous congestion in chronic HF due to endothelial damage and disruption [68,70]. It has been associated with poor outcomes in patients with HF and reduced ejection fraction [73]. In one prospective study by Jukneviciene et al. on patients with acute dyspnea admitted to the emergency department, CD146 strongly correlated with the degree of vascular and tissue congestion assessed imagistically, regardless of the NT-proBNP levels [74]. Although a promising biomarker, it must be further validated in studies with different HF clinical scenarios.

Another peptide that maintains vascular integrity through the barrier stabilization of the endothelial cells is adrenomedullin (ADM), which has been proposed as a potential biomarker for congestion in patients with HF [75,76,77]. In additional studies involving patients hospitalized for AHF, ADM levels exhibited a significant association with clinical congestion upon admission and residual congestion before discharge [77,78]. Furthermore, various other investigations have demonstrated a correlation between ADM levels and clinical congestion, as well as with mean pulmonary artery pressure and pulmonary capillary wedge pressure [79].

The ST2 (suppression of tumorigenicity 2) protein is significantly associated with HF. Elevated levels of sST2 have been observed in the bloodstream of HF patients compared to healthy individuals [80]. ST2 is involved in the pathophysiology of HF, particularly in promoting myocardial fibrosis and ventricular remodeling. Additionally, it serves as a valuable biomarker for prognosis and risk stratification in patients with HF [81,82]. Monitoring sST2 levels can provide valuable information for managing and predicting outcomes, as sST2 positively correlates with echocardiographic indicators of right-sided HF and invasively measured central venous pressure in patients with AHF. Furthermore, sST2 is a surrogate marker of diuretic resistance for HF patients with renal dysfunction at presentation [83,84,85]. It is also associated with an increased risk of mortality and hospitalization [86,87].

Endothelin-1 (ET-1) is a potent vasoconstrictor with additional anti-natriuretic and mitogenic properties [68]. It plays a crucial role in modulating salt and water homeostasis and maintaining vascular tone and blood pressure in healthy individuals. Furthermore, ET-1 contributes to inflammation and neurohormonal activation. Elevated ET-1 levels increase peripheral vascular resistance, exacerbating hypertension and impairing cardiac function [88,89]. In patients with HF with preserved ejection fraction (HFpEF), circulating levels of ET-1 are higher and strongly correlated with mean pulmonary artery pressure and pulmonary capillary wedge pressure [90,91]. ET-1 has been investigated both as a predictor of HF and as a prognostic marker in patients with established acute or chronic congestive HF. Higher baseline ET-1 concentrations are independently associated with worse clinical outcomes and a more rapid decline in kidney function. Notably, treatment with dapagliflozin has shown benefits across a range of ET-1 concentrations, leading to a modest decrease in serum ET-1 concentration [92].

Although these emerging congestion biomarkers are very promising, some still need validation. On the other hand, several other markers commonly encountered in daily practice can be used to estimate congestion, such as hemoglobin, hematocrit, blood urea nitrogen, serum protein, and liver enzymes [4,5,69,93,94]. For example, by using hemoglobin and hematocrit, one can quickly assess the change in plasma volume through the Duarte formula [95]. A threshold of >5.5 mL/g for the estimated plasma volume has been linked to excessive volume overload and poor outcomes [96]. Another useful parameter is the increase in the creatinine level during diuretic therapy, which indicates hemoconcentration through successful decongestion rather than worsening renal function [68,97].

### 5.3. Imaging Methods for Assessing Congestion

When considering the visual assessment of congestion, the most potent imaging tool in evaluating congestion starting from the pre-hospital setting to the emergency department, in-hospital, and ambulatory ward is ultrasonography (lung ultrasound and abdominal inferior vena cava measurement) [5]. With lung ultrasound, the rapid determination of B-lines (ultrasound lung comets “comet tail”-like) at the patient’s bedside is possible. The number of B-lines is proportional to congestion severity, with good sensitivity and specificity [98]. Lung ultrasound can assess residual pulmonary congestion pre-discharge and is considered a strong predictor of adverse outcomes post-discharge [99]. A dilated inferior vena cava (IVC) with reduced respiratory variations (<50%) is a strong predictor of elevated right atrial pressures and systemic congestion [100,101].

Renal Doppler venous flow offers valuable insights into the hemodynamic status and renal function, particularly in patients with congestive HF. Changes in renal venous flow patterns can transition from a continuous to a discontinuous pattern, indicating alterations in renal perfusion and congestion [102,103].

## 6. Congestion Management: Are Diuretics the Sole Remedy for Congestion Relief?

The primary focus of therapy for HFrEF and HFpEF is decongestion, given their comparable clinical profiles of congestion during AHF decompensation. The significant correlation between the early administration of intravenous (IV) loop diuretics and reduced in-hospital mortality underscores the endorsement of this approach as first-line therapy in AHF (Class I, level of evidence B) [58]. The early initiation of loop diuretics improves dyspnea substantially within 6 h. The main mechanism of action is renal natriuresis and diuresis. The diuretic response is influenced by factors such as the type and dosage of the diuretic, the extent of volume overload, body composition, and kidney function. Indicators of a good response to diuretics are weight loss, fluid output, and urinary Na [4]. The absence of clinical congestion and the lack of signs or symptoms, including dyspnea at discharge, are deemed inadequate predictors for achieving complete decongestion. Thus, patients still experience high rates of hospital readmissions, with only one-third remaining congestion-free at 60 days [39,41]. High levels of natriuretic peptides at discharge, despite symptom relief, indicate treatment failure and are considered one of the most robust predictors of mortality [104]. This underscores the lack of correlation between weight loss and decongestion [3,35].

The main goal for patients admitted with acute decompensated HF is to prevent residual decongestion. To do so, one must thoroughly assess the patient on admission for possible features that might lead to residual congestion. For example, it is important to distinguish from the beginning what the mechanism responsible for congestion is: volume overload or redistribution. Also, the presence of kidney impairment, liver failure, hypoproteinemia, increased intra-abdominal pressure, or low blood pressure may alter the diuretic response and impede decongestion. Urinary output, urinary ionogram, weight change, NT-proBNP monitoring, and congestion scores are all helpful tools in monitoring the decongestion response and thus guiding the dose escalation of loop diuretics or bail-out therapy through ultrafiltration [4,35,97,105]. The chronic use of loop diuretics in patients with chronic HF might lead to diuretic resistance, impeding decongestion in the case of a decompensation episode [106]. The diuretic-induced hypertrophy of distal tubular renal cells explains this phenomenon, causing compensatory Na reabsorption and thereby reducing natriuresis [107].

According to the position statement of the Heart Failure Association of the European Society of Cardiology (HFA ESC) [4], one efficient way to tackle residual decongestion is to evaluate early diuretic response in the event of an episode of AHF. This is conducted by determining urinary output and measuring urinary Na excretion at 2 h from the initial dose of loop diuretics. If a urinary spot Na < 50–70 mEq is measured at 2 h and the urinary output is less than 150 mL per hour, then it is safe to say that there is an insufficient diuretic response, and doubling the dose of IV loop diuretics should be considered [4]. Certainly, this process continues until the maximum dose is reached, if natriuresis and urinary output remain insufficient. However, one should remember that loop diuretics should be administered in a protein-bound form and dosed based on protein plasma levels for adequate secretion in the proximal tubule.

Diminished plasma protein levels resulting from chronic loss or reduced production impede plasma refill from the interstitium and, thus, the delivery of loop diuretics to the kidney [108]. In this condition, escalating the loop diuretic dose before correction for hypoproteinemia is futile. Another aspect that needs to be taken into consideration when tackling residual decongestion, especially in patients with advanced HF, is hypochloremia. Hypochloremia is a marker of bad prognosis, promotes Na retention, and is associated with diuretic resistance [109]. Rubin Albert et al. first observed this phenomenon in the 1950s, noting that patients undergoing treatment with mercurial diuretics experienced hypochloremia and diuretic resistance. However, administering exogenous lysine chloride successfully addressed this resistance, restoring the diuretic response [110]. Low chloride (Cl) levels stimulate renin secretion and the up-regulation of NaCl channels in the distal tubule, thus increasing Na reabsorption [111]. These mechanism insights led to administering hypertonic saline with loop diuretics to augment diuresis [112,113]. Theoretically, hypertonic saline alone can restore low chloride levels, prevent Na retention, osmotically shift fluid into the intravascular compartment, temporarily reduce neurohormonal activation, and thus improve diuresis [114]. However, a potential issue arises due to tubuloglomerular feedback, which could eventually decrease renal blood flow [114]. This can be overcome with the concomitant administration of furosemide, thus potentiating diuresis. This was tested in several studies, where hypertonic saline was administered alongside high-dose intravenous furosemide, as opposed to standard therapy, in patients with acute decompensated HF [112,113]. The results were a higher effective diuresis, a shorter hospital stay, symptom improvement, and reduced hospital readmission [112,113]. Further randomized controlled studies are required to elucidate and validate this therapeutic regimen. Nevertheless, it is an option for patients who have not responded to conventional therapies and are hyponatremic.

Checking plasma and urinary electrolytes, urinary output, plasma, and urinary proteins should be conducted as soon as possible in the early phase of admission, as it could impact the diuretic response and avoid residual decongestion. As mentioned earlier, more than half of HF patients exhibit little or no change in body weight before admission, and these patients are likely to experience fluid redistribution. It is unwise to try escalating diuretic doses in these patients because it can only further decrease plasma volume, reduce renal blood flow, enhance neurohormonal activation, and worsen renal failure. The main goal in this population is to improve venous capacitance and reduce cardiac filling pressures [58]. Adding vasodilators to low doses of diuretics can achieve this by reducing preload and inducing arterial vasodilation [115].

In certain cases, achieving decongestion can be challenging, especially when factors such as hypotension, severe systolic dysfunction, or severe pulmonary hypertension hinder the process. Consequently, the utilization of inotropic agents or heart rate reduction may be warranted to overcome these challenges. Numerous pharmacological agents are being tested to improve pulmonary capillary wedge pressure or venous capacitance, but besides symptomatology, they have failed to improve outcomes (Figure 4) [115]. If the patient is hemodynamically stable, the early introduction of neurohormonal blockers such as angiotensin receptor–neprilysin inhibitors (ARNis), sodium–glucose cotransporter two inhibitors (iSGLTs), mineralocorticoid receptor antagonists (MRA), and beta-blockers might reduce cardiac filling pressures and improve venous capacitance and fluid redistribution [58]. This strategy has been tested in the STRONG-HF trial. An intensive treatment strategy of the rapid up-titration of guideline-directed medication (at least two neurohormonal blockers) in stable hospitalized patients can overcome residual decongestion and improve outcomes [116].

Diuretic resistance due to distal tubule Na avidity can be overcome by using thiazide or thiazide-like diuretics. These agents block the sodium-chloride cotransporter from the distal tubule, promoting natriuresis and kaliuresis [117,118]. They enhance diuresis when used in combination with loop diuretics. However, there are some significant pitfalls when using these agents: they cannot dilute urine, necessitate protein binding for tubular secretion, and are independent predictors of hyponatremia and hypokalemia, correlating with increased all-cause mortality [118,119]. Using mineralocorticoid receptor blockers can counteract thiazide-induced hypokalemia. Due to their anti-neurohormonal activity, mineralocorticoid antagonists can prevent congestion; otherwise, they are not helpful for decongestion therapy in an acute setting. The ATHENA-HF trial failed to show the superiority of high-dose spironolactone compared to the lowest dose in reducing NT-proBNP and increasing urinary output in patients with decompensated HF [120,121].

Acetazolamide is a tempting, new, or otherwise old agent used to facilitate congestion in patients with decompensated HF. It is a carbonic anhydrase inhibitor that facilitates Na reabsorption in the proximal tubule and is formally known for treating altitude pulmonary edema and glaucoma [3,4]. As 65% of Na is absorbed in the proximal tubule, inhibiting its absorption at this stage through the carbonic anhydrase inhibitor will lead to increased natriuresis and the enhanced delivery of chloride to the macula densa, resulting in reduced neurohormonal activity [16]. It is only logical to use this agent to enhance diuresis and decongestion to achieve euvolemia. Mullens et al. evaluated acetazolamide in the ADVOR study, which involved 519 patients with acute decompensated HF. The study demonstrated that adding acetazolamide to loop diuretics in these patients led to more effective decongestion, with impacts and side effects comparable to those observed in the placebo group [106]. However, there are some drawbacks to this strategy. The addition of intravascular acetazolamide to IV furosemide, aimed at achieving more efficient decongestion, did not result in a reduction in all-cause mortality or rehospitalization for HF, which remained similar between groups (29% vs. 27%) [122,123]. Despite observing a greater increase in natriuresis and urinary output in patients with a glomerular filtration rate (GFR) < 40 mL/min/m^2^ from the acetazolamide group, this was offset by a higher incidence of worsening renal function, whether this primarily indicates positive clinical decongestion or represents an early sign of acute kidney injury [122,123]. Overall, the adage of “better sooner than later” does not seem to apply to the prognosis of HF with this drug, necessitating further investigation through additional studies.

The new agent that revolutionized treatment in patients with chronic HF in recent years regarding survival and rehospitalization was SGLT2 inhibitors [124,125]. Numerous proposed mechanisms were responsible for their beneficial effects, one of them being natriuresis, which led clinicians to test this drug in patients with AHF in addition to loop diuretics to increase diuresis, relieve congestion, and prevent residual decongestion [126]. The most extensive trial to test SGLT2 inhibitors in AHF was the EMPULSE trial, where patients were assigned to receive empagliflozin 10 mg daily or placebo [127]. The trial concluded that initiating empagliflozin in patients with AHF is safe and well tolerated regardless of the ejection fraction and resulted in a clinical benefit at 90-day follow-up in terms of all-cause mortality and time to the first HF event [127]. However, it is essential to note that the treatment was randomized at a median time of three days from hospitalization when patients were considered stable [127]. Thus came the DICTATE-HF trial, where researchers tested the diuretic efficacy of SGLTi when initiated within 24 h of hospital admission in patients with AHF [128]. The study failed to show a statistically improved diuretic response at five days and discharge for dapagliflozin compared to standard usual care [112]. The effectiveness of SGLTi as a diuretic regime in addition to standard decongestive therapy in patients with acute decompensated HF was tested in the EMPAG-HF randomized study [129]. The study showed that empagliflozin-treated patients, compared to those with standard treatment, experienced a 25% increase in 5-day urine output without kidney injury and a greater decline in natriuretic peptide levels [129]. Conversely, the EMPAG-HF study underscored the effectiveness of SGLT2 inhibitors as adjunctive diuretic therapy alongside standard decongestive treatment in patients with AHF. The results indicated that patients treated with empagliflozin experienced a 25% increase in 5-day urine output without kidney injury and a more significant reduction in natriuretic peptide levels than those receiving standard care [129].

In summary, the usage of SGLT2 inhibitors in AHF needs to be better established, particularly in alleviating congestion. Potential risks, such as ketoacidosis or acute kidney injury, need to be carefully considered, especially during the “acute vulnerable phase”, characterized by elevated lactate or inflammatory levels, hemodynamic instability, or low blood pressure [130,131]. Nonetheless, initiating SGLT2 inhibitors early during HF admission offers advantages regarding short-term outcomes and tackling residual congestion.

## 7. Conclusions

Gaining insight into the pathophysiology and clinical presentation of congestion is essential for improving the management of HF patients. Utilizing straightforward tools like pulmonary echocardiographic parameters to assess cardiac filling pressure or circulating serum congestion biomarkers, clinicians can anticipate future episodes of congestion decompensation or identify residual congestion following an AHF episode. This multiparametric approach for assessing congestion might be especially beneficial for monitoring and guiding decongestive therapies.

## Figures and Tables

**Figure 1 biomedicines-12-00951-f001:**
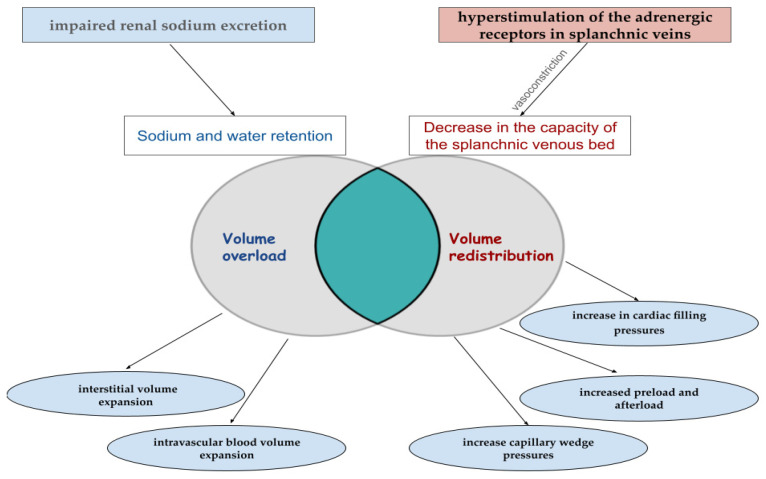
Schematic representation illustrating the interplay between volume overload and volume redistribution in congestion.

**Figure 2 biomedicines-12-00951-f002:**
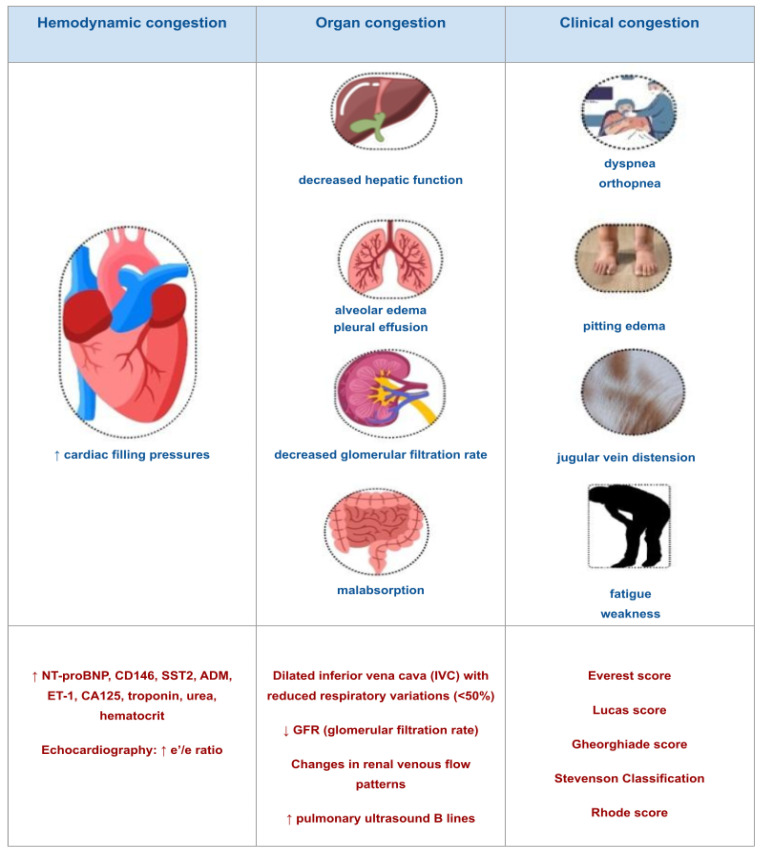
An overview of hemodynamic, organ, and clinical congestion with diagnostic tools. This figure presents a comprehensive view of hemodynamic, organ, and clinical congestion, illustrating their interrelation in congestive states. Several diagnostic tools, such as biomarkers, echocardiography, lung ultrasound, vena cava dimension, and various scoring systems for assessing congestion, are illustrated around each type of congestion. This visual representation elucidates the multifaceted approach to detect and assess congestion, highlighting the importance of integrating multiple diagnostic modalities for comprehensive evaluation and management. Abbreviations: NT-proBNP = N-terminal pro-brain natriuretic peptide. CD146 = cluster of differentiation 146; SST2 = Soluble suppression of tumorigenicity 2. ADM = Adrenomedullin. ET-1 = Endothelin 1. CA125 = cancer antigen 125. ↑ = increase. ↓ = decrease.

**Figure 3 biomedicines-12-00951-f003:**
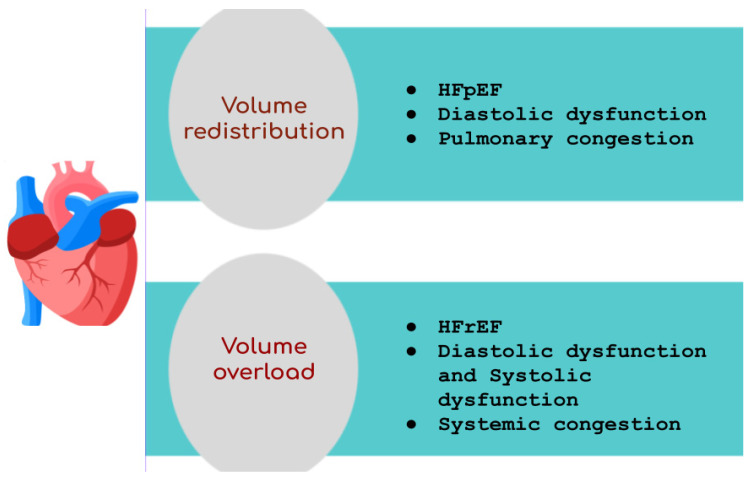
Phenotypes of heart failure. Diastolic dysfunction, characterizing HF with preserved ejection fraction (HFpEF), leads to pulmonary congestion through volume redistribution. HF with reduced ejection fraction (HFrEF) entails volume overload, resulting in systemic congestion, and involves both systolic and diastolic dysfunction.

**Figure 4 biomedicines-12-00951-f004:**
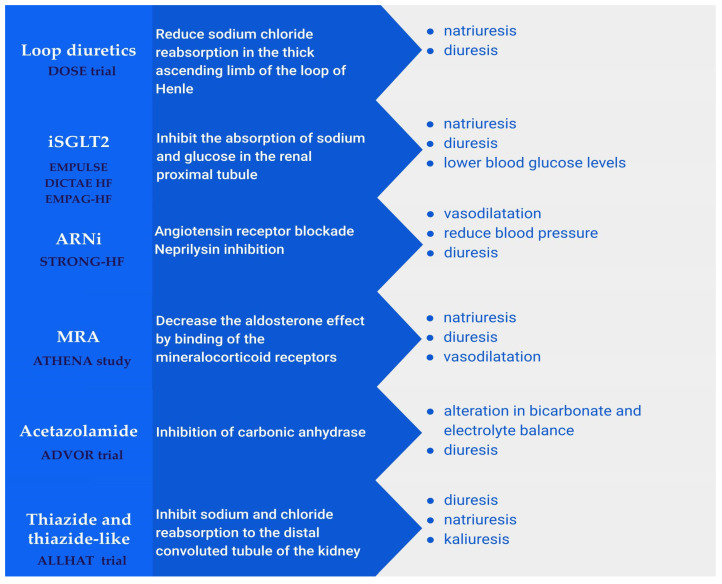
Therapeutic strategy in heart failure. A concise description of drugs, their mechanisms of action, and their effects. Additionally, a representative trial for each drug is presented.

**Table 1 biomedicines-12-00951-t001:** Clinical congestion scores. Abbreviations: JVP = jugular venous pressure. NT-proBNP = N-terminal pro-brain natriuretic peptide. BNP = brain natriuretic peptide.

**Clinical** **congestion scores**	**Name of the Score**	**Roles and Utility of the Score**
	Everest score	Evaluation of dyspnea, orthopnea, jugular venous distension, rales, edema, fatigue
	Lucas score	Evaluation of orthopnea, external jugular vein distension, pitting edema, the dose of diuretics during the past week, weight gain since the previous clinic visit
	Gheorghiade score	Evaluation of orthopnea, JVD, edema, hepatomegaly, orthostatic testing, 6 min walk test, Valsalva maneuver, BNP, NT-proBNP
	Stevenson Classification	The 4 Stevenson profiles:
		Profile A, patients with no evidence of congestion or hypoperfusion (dry-warm); Profile B, congestion with adequate perfusion (wet-warm); Profile C, congestion and hypoperfusion (wet-cold); and Profile L, hypoperfusion without congestion (dry-cold)
	Rhode score	Evaluation of orthopnea, JVD, rales, edema, third heart sound

**Table 2 biomedicines-12-00951-t002:** Novel biomarkers of congestion. Abbreviations: CA125 = cancer antigen 125. CD146 = cluster of differentiation 146. SST2 = Soluble suppression of tumorigenicity 2. ADM = Adrenomedullin. ET-1 = Endothelin 1. HF = heart failure. RV = right ventricle.

**New biomarkers of congestion**	**Name of the Biomarker**	**The Main Characteristics of the Biomarker**
	CA 125	Localization: serosal epithelial cells released in the circulation consequence of mechanical stress induced by elevated hydrostatic pressure Roles: congestion marker ovarian cancer marker detect patients with RV failure and chronic fluid overload predict both short and long-term outcomes in decompensated HF potential to guide diuretic therapy
	CD 146	Localization: junctions of endothelial cells throughout the human vascular system, smooth muscle cells, and pericytes released in circulation consequence of endothelial dysfunction, vascular injury, or mechanical vascular stretch Roles: angiogenesis, vessel permeability, and leukocyte transmigration reflect vascular and tissue congestion
	ADM	Roles: vascular integrity by the barrier stabilization of the endothelial cells biomarker for congestion in patients with HF correlate with clinical congestion, mean pulmonary artery pressure, and pulmonary capillary wedge pressure
	SST2	Roles: promoting myocardial fibrosis and ventricular remodeling prognosis and risk stratification in patients with HF associated with an increased risk of mortality and hospitalization
	ET-1	Roles: roles in inflammation anti-natriuretic and mitogenic properties vasoconstrictor, increase peripheral vascular resistance, exacerbate hypertension, and impair cardiac function predictor of HF

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
