# Peer review of "The Congestion “Pandemic” in Acute Heart Failure Patients"

_biomedicines, 2024, doi:10.3390/biomedicines12050951_

Round 1
Reviewer 1 Report
Comments and Suggestions for Authors
This a very nice review and the approach to heart failure is commendable. I really do not have anything to ad to the paper. It is well written and accurate with good references.
Author Response
Response to Reviewer 1
Thank you very much for taking the time to read our manuscript. We are grateful for your generous feedback.
Reviewer 2 Report
Comments and Suggestions for Authors
In this review, the authors summarize the pathophysiology, diagnosis, and management of congestion in acute heart failure. The topic is interesting but the manuscript has several limitations.
COMMENTS/QUERIES
1.Title: “The congestion “Pandemic” in Acute Heart Failure Patients”.
Reviewer
What is this title supposed to mean?
2. Understanding congestion: Volume overload or volume redistribution. “Congestion is manifested clinically through signs and symptoms due to extracellular fluid accumulation due to increased left-sided cardiac filling pressures [3,4]. The increase in cardiac filling pressures is an early indicator of hemodynamic congestion, which precedes the development of congestive symptoms in days or weeks [5]”.
Reviewer
Understanding the physiology of congestion requires conceptualizing intravascular volume into functional compartments, consisting of the unstressed and stressed blood volumes. The venous capacitance vessels function as a key reservoir of unstressed blood volume and are highly vasoactive and innervated by sympathetic nerve fibers, allowing for vasoconstriction to recruit blood from unstressed pools to the central circulation in times of increased demand. In patients with HF, fluid accumulation and redistribution frequently coexist, as these evolutionary adaptations are engaged by variable degrees of neurohormonal activation, confounding interpretation of venous pressures as a metric of TBV, much less total body fluid volume
3. Figure 1. - Overview of Hemodynamic, Organ, and Clinical Congestion with Diagnostic Tools.
Reviewer
The increase in cardiac filling pressures and venous pressure may not be early events in congestion development. Volume accumulation often precedes intravascular pressure elevation and pressure does not necessarily reflect total body volume (see also previous comment). Both central venous pressure (CVP) and pulmonary capillary wedge pressure (PCWP) are determined by multiple forces the most important being the systolic and diastolic function of the heart. As a result, two patients with similar intravascular volume may have dramatically different CVPs if the preceding forces differ, whereas, patients with the similar CVP may have very different plasma volumes. Modulation of vascular tone, by either vasoconstrictors or vasodilators, may lead to marked changes in CVP within the same individual at constant blood volume.
Comments on the Quality of English LanguageModerate editing
Author Response
Response to Reviewer 2
Thank you very much for taking the time to read our manuscript. We appreciate your review and comments on our manuscript.
We selected this title because during the COVID-19 pandemic, as we planned to write about congestion in heart failure, we aimed to highlight the substantial number of heart failure patients, akin to pandemic proportions. Moreover, the title's engaging nature was intended to garner interest in the article, leading to our manuscript being invited for submission to the Biomedicines journal. We are open to changing the title if you believe another one would be more suitable. One suggestion could be: The Congestion Phenomenon among patients with acute heart failure - a Narrative Review.
In the manuscript, we attempted to delineate the two mechanisms of congestion separately while acknowledging their coexistence. “ The study highlights that fluid redistribution and accumulation coexist [49]. We mentioned in the manuscript th physiology of congestion. We also made some modifications to Figure 2 (Overview of Hemodynamic, Organ, and Clinical Congestion with Diagnostic Tools.)
- The first stage is hemodynamic congestion, characterized by the elevation of cardiac filling pressures and the increase in venous pressures without clinical manifestation [37,38]. Subsequently, organ congestion is caused by redistribution and accumulation of fluid within the extracellular and third space, and clinical congestion appears [37].
- Volume redistribution was assessed in three major trials: COMPASS-HF, HOMEOSTASIS–HF, and CHAMPION by invasive monitoring of hemodynamics. All findings consistently indicate that weight gain resulting from volume overload does not typically precede an episode of acute decompensation in most patients; instead, there is an observed increase in cardiac filling pressures [9-11]. HF is a state of neurohormonal activation, inflammation, and endothelial dysfunction that promotes veno-arterial constriction. This leads to a subsequent decrease in venous bed capacitance with consequent blood volume redistribution and increased preload and afterload
- Venous distension augments endothelial dysfunction and sympathetic activation, further promoting the fall in splanchnic venous capacitance and consequent fluid redistribution [15].
- Hyperstimulation of the α1 and α2 adrenergic receptors in splanchnic veins characterizes the mechanism, increasing cardiac filling and capillary wedge pressures [12]
Reviewer 3 Report
Comments and Suggestions for Authors
Dear
Biomedicines
Editorial Office
I have reviewed and read the article titled "The congestion “Pandemic” in Acute Heart Failure Patients" by the authors Daniela Mocan, Radu I. Lala, Maria Puschita, Luminita Pilat, Dan A. Darabantiu and Adina Pop-Moldovan.
The review is interesting, well written, and reviews the recent literature on heart failure. However, I have some observations that could help the authors in their revision and that I point out below.
Abstract
page 1 line 13 adds the abbreviation for acute heart failure, and for heart failure.
In an article, when an abbreviation is introduced, it is placed in the first mention and then throughout the manuscript it is used. In this sense, the authors use or do not use it throughout the entire document. Can the authors please approve this? Throughout the document, below I point out an example of this:
Page 1, lines 31 and 36. Page 2, lines 51. page 3, lines 109,110 and 130. Page 4, lines 164,165,173 and 175. page 6, lines 217 and 223, page 7, lines 269 and pages 8 lines 308 and 312 .This is just an example for heart failure and acute heart failure, since the authors sometimes use the abbreviation and in other cases not.
The two figures proposed by the authors are not mentioned in the text, please mention them in the manuscript in the corresponding section.
in the section 2. Understanding congestion: Volume overload or volume redistribution.
The authors could divide it into the two subsections that mention volume averload and volume redistribution
page 3 line 34 what types of adrenergic receptors are involved, specify.
heart failure manifests a strong avidity for Na is a subheading of section 2, add it as such and not in the sentence as a full stop
Sodium in a cation because in some places in the manuscript they use the symbol alone, or with the superscript + or followed by the symbol plus the superscript + followed by cation, please homologate this throughout the entire manuscript.
in this section 4. Does heart failure phenotype predict congestion mechanism?
Could the authors add a figure that summarizes this section?
page 4 line 215 when the construct "on the other hand" is used, it is used after a full stop, because one is going to change one's mind and not after a full stop
in section 5. Clinical and paraclinical integrative assessment of congestion
The authors could add a table that summarizes the classifications that exist.
in section 5.2. The new congestion biomarkers on the horizon
The authors could add a table that summarizes the biomarkers that exist
in section 6. Congestion management: Diuretics the sole remedy for congestion relief?
Could the authors add a table or figure summarizing this section?
page 11 line 500 please delete the sentence
The new kid on the block is very colloquial, a scientific review
Thank you very much for allowing me to carefully review the manuscript, the reviewer
Comments on the Quality of English LanguageOverall the article is well written but requires some minor revision.
Author Response
Response to Reviewer 3
Thank you very much for dedicating your time to review our manuscript. We highly value your observations, finding them extremely beneficial. We have addressed all the corrections you suggested, including the incorporation of abbreviations for heart failure, acute heart failure, and sodium as per your recommendation. Additionally, we have created two new figures and two tables, integrating them along with the others into the text according to your guidance. Furthermore, we have included the suggested subsections.
Round 2
Reviewer 2 Report
Comments and Suggestions for Authors
My concerns remain.
Comments on the Quality of English LanguageMinor editing
Author Response
We appreciate and thank you for your feedback.
Our manuscript on the pathophysiology of congestion does not attempt to provide a detailed explanation of its complexity, but rather to underscore it. Our objective is to draw clinicians' attention to the primary mechanisms—hemodynamic congestion and clinical congestion—and to advocate for the use of available clinical tools and pharmacological interventions to prevent decompensation, as ultimately everything culminates in decompensation.